# The Dynamic Interplay between Loss of Semantic Memory and Semantic Learning Capacity: Insight from Neologisms Learning in Semantic Variant Primary Progressive Aphasia

**DOI:** 10.3390/brainsci13050788

**Published:** 2023-05-11

**Authors:** Simona Luzzi, Sara Baldinelli, Chiara Fiori, Mauro Morelli, Guido Gainotti

**Affiliations:** 1Department of Experimental and Clinical Medicine, Polytechnic University of Marche, 60126 Ancona, Italy; 2Neurologic Clinic, Azienda Ospedaliero-Universitaria delle Marche, 60126 Ancona, Italy; 3UO Neurologia, AST Ancona, 60019 Senigallia, Italy; 4Institute of Neurology, Università Cattolica del Sacro Cuore, 00168 Rome, Italy

**Keywords:** semantic variant PPA, semantic learning tasks, neologisms, new conceptual representations, left temporal lobe

## Abstract

Semantic Variant of Primary Progressive Aphasia (svPPA) has often been considered as a loss of knowledge stored in semantic memory, but might also be due to a general disruption of mechanisms allowing the acquisition, storage, and retrieval of semantic memories. In order to assess any parallelism in svPPA patients between loss of semantic knowledge and inability to acquire new semantic information, we administered a battery of semantic learning tasks to healthy individuals and svPPA patients, where they were requested to learn new conceptual representations and new word forms, and to associate the former with the latter. A strong relation was found between loss of semantic knowledge and disruption of semantic learning: (a) patients with severe svPPA had the lowest scores in the semantic learning tasks; (b) significant correlations were found between scores obtained in semantic learning tasks and scores obtained in semantic memory disorders in svPPA patients.

## 1. Introduction

Anomia and semantic-lexical disorders in aphasic stroke patients have been mainly discussed in the context of the distinction between “access” and “storage” disorders. [1,2]. According to this model, in storage disorders, the semantic representations are damaged or lost, whereas in access disorders, the representations are intact but cannot be easily accessed. However, if both instances of access and storage disorders have been repeatedly described in vascular aphasic patients [3,4,5], a loss or degradation of semantic knowledge has been much more frequently reported in patients with semantic disorders resulting from degenerative brain diseases, namely, in Alzheimer’s disease (AD) (e.g., [6,7]) and Semantic variant of Primary Progressive Aphasia (svPPA) (e.g., [8,9,10] in association with left anterior temporal damage). This is particularly true of svPPA because, according to the criteria proposed by Gorno-Tempini et al. [11], impaired confrontation naming and impaired single-word comprehension are the core features of this variant of PPA, and because non-verbal semantic disorders (impaired object knowledge) are also frequently reported. These non-verbal semantic disorders can even be in the foreground when the brain atrophy prevails in the right hemisphere [12,13,14,15]; this is the reason why svPPA is also frequently called Semantic Dementia. On the other hand, semantic disorders are not considered as core features of the other two variants of PPA, the Non-Fluent and the Logopenic PPA. Apraxia of speech and/or agrammatism in language production are the core features of the Non-Fluent variant [11,16], whereas impaired single-word retrieval in spontaneous speech and naming and impaired repetition of sentences and phrases are the key features of the Logopenic PPA. Furthermore, the significant word-finding problems potentially found in the Logopenic variant are due to lexical retrieval, and not to properly semantic problems [11,17]. However, as lvPPA progresses, atrophy spreads to temporal areas associated with lexical-semantic and conceptual processing [18] and semantic impairment may appear in lvPPA [19]. The semantic disturbances observed in vascular aphasic patients are more similar to the language disorders shown by svPPA patients, but again, the mechanisms underlying these apparently similar disorders partially differ. For instance, Jefferies and Lambon Ralph [10] compared the semantic disorders of two groups of patients with stroke aphasia and semantic dementia, respectively, and showed that patients with stroke aphasia had inconsistent results in tasks requiring different kinds of semantic processing, whereas SD patients showed strong correlations between different semantic tasks, regardless of input/output modality. Furthermore, a substantial consistency was found in these patients when a set of items was assessed several times. The semantic disorders observed in AD patients are much more similar from the standpoint of cognitive neuropsychology, and are informative for clarifying the structure of semantic memory because, according to many authors (e.g., [20,21,22]), semantic deficits in AD could result from the degradation of core semantic representations. The interactions existing between episodic and semantic memory [23] and the partly different memory disorders shown by AD and svPPA patients [24] suggest, however, that a different mechanism may underlie the storage disorders of these different degenerative brain diseases. In fact, it is usually assumed that AD patients cannot insert new information into episodic memory due to a disruption of mechanisms subsumed by the hippocampus and entorhinal cortex, whereas svPPA patients show a loss of semantic knowledge due to the atrophy of the anterior temporal cortex, where conceptual representations are stored [25]. An objection raised to those interpretations assuming a stable loss of semantic knowledge in svPPA has been that they are based on a rather static model of formation and storage of conceptual representations; this objection has been strengthened by the results of studies evaluating the capacity of patients with severe semantic disorders to learn new words and new semantic representations.

The influence of learning capacity on the semantic rehabilitation in SD patients was originally suggested by Swales and Johnson [26], who showed that a patient with semantic memory loss was able to relearn (with considerable repetition) some lost semantic information. However, a qualitative analysis of this re-learning process suggested that it could be based on the interaction between semantic and episodic memory mechanisms [23], rather than on strictly semantic processes. This debate could be fueled by the observation that a semantic learning capacity was also documented in the context of stroke aphasia rehabilitation by several authors (e.g., [27,28,29,30,31,32,33,34,35,36,37,38,39,40]) who investigated the variables that can facilitate the relearning or novel word learning in people with aphasia. Some of these studies (e.g., [33]) faced this problem by pairing novel phonological word forms (i.e., neologisms) with existing semantic representations (i.e., known meanings). Other investigations (e.g., [29,34,35]) paired familiar word forms (i.e., known words) with new semantic representations (i.e., novel semantics) or new semantic representations with neologisms [36]. These investigations have consistently shown that individuals with chronic post-stroke aphasia are able to learn novel words, but also that their performance is variable and influenced by lexical-semantic factors, because individuals with impaired lexical-semantic processing demonstrate poorer novel word learning when compared to people with intact lexical-semantic processing. These results might suggest that verbal and non-verbal disorders in SD patients could be due not only to a loss of semantic knowledge (e.g., [12,41]), but also to a general defect related to the learning and storing of semantic knowledge. As a matter of fact, the question of verifying if the semantic system of SD patients is able to learn new concepts has never (to our knowledge) been addressed, nor can be inferred from the patients’ clinical history. Graham et al. [27,28] have shown, indeed, that SD patients can relearn “forgotten vocabulary” through frequent practice, but the benefit of this rote learning is quickly lost once practice ceases and, in any case, these patients did not learn new concepts.

For this reason, in the present investigations, we studied the semantic learning capacity of SD patients using a methodology similar to the one employed by Digman et al. [36] in their studies on novel word learning in patients with post-stroke aphasia by pairing novel word forms (i.e., neologisms) with new conceptual-semantic representations. In the present experimental setting, these new conceptual-semantic representations were formed by fantasy geometric shapes, whereas neologisms were anagrams of Italian words plausible with the Italian phonological rules.

We predicted that, if a general defect of semantic acquisition, storage, and preservation is responsible for the loss of knowledge observed in svPPA patients, then a relation should exist between their semantic defect, documented in standard neuropsychological tasks, and their capacity to learn new semantic information. In our study, we also tried to take into account the characteristics of semantic learning tasks that could increase the learning difficulty and allow predicting the severity of the semantic breakdown shown by svPPA patients. We considered that two features should increase this difficulty: namely the abstract vs. concrete nature of the stimuli and the arbitrariness of the links existing between new conceptual representations and novel word forms. We, therefore, constructed a “Semantic Learning Experimental Battery,” where we tried to include tasks with different levels of stimulus concreteness and arbitrariness between new conceptual representations and novel word forms, or between different components of the new conceptual representations.

## 2. Materials and Methods

### 2.1. Participants

A total of 12 patients (9 males, 3 females) with the semantic variant of Primary Progressive Aphasia (svPPA) and 6 normal individuals (4 males, 2 females) were enrolled in the study. Patients with svPPA were included in the study if the following criteria were present:-Diagnosis of Primary Progressive Aphasia—semantic variant according to the current criteria [11];-Clinical history characterized by predominant and early anomia and language comprehension problems;-Predominant anterior left temporal atrophy detected by an MRI and/or predominant anterior left temporal hypoperfusion at the FDG-PET scan; potential participants who had anterior temporal atrophy or hypoperfusion that was comparable across the hemispheres were, however, also included in the study.-Absence of behaviocral problems that could interfere with adherence to the experimental battery.

Participant confidentiality precludes public archiving of the data. The data may be accessed upon request to the Scientific Committee of the Neurology Clinic at Marche Polytechnic University (s.luzzi@univpm.it). Interested readers will be required to fill a “Collaboration Statement.”

Inclusion criteria for normal individuals were as follows:-Absence of neurological diseases, psychiatric problems, and systemic diseases (i.e., liver failure etc.) that could alter cognition. Partners and relatives of patients were not allowed to be part of the experimental battery as normal controls to avoid interference.

Diagnosis was based on the patients’ clinical history, neurological, and neuropsychological examination, and supported by structural (MRI scan) and functional (PET scan) imaging.

The mean age of svPPA patients was 67.33 years old (SD 6.30), mean education duration was 11.91 years (SD 4.83).

They were classified at the early (8 patients) or moderate stage (4 patients) of the disease, depending on the magnitude of their semantic breakdown, according to Julien et al. [42]. The mean illness duration was 2.4 years (SD 0.64; range 1–3 years) in mild SD, and 5.5 years (SD 0.58; range 5–6 years) in moderate SD. Early SD patients were 6 men and 2 women with a mean age of 65.12 (SD 6.65) and a mean education duration of 12.5 years (SD 4.84). Patients with moderate SD were 3 men and 1 woman with a mean age of 71.75 (SD 6.5) and a mean education duration of 10.75 years (SD 5.31). The mean age of controls was 66.16 years old (SD 6.85) and their mean education duration 10.66 years (SD 4.41).

Demographic variables did not differ both in the comparison between controls and SD patients: age (t = 0.36; *p* = 0.72), education duration (t = 0.53; *p* = 0.60), sex (χ2 = 0.14; df = 2; *p* = 0.71); and in the comparison between patients with mild and moderate svPPA: age (t = 1.9; *p* = 0.8), education (t = 0.01; *p* = 0.98), sex (χ2 = 0.0; df = 2; *p* = 1). All participants provided informed written consent to participate in the study, which was approved by the local Ethics Committee.

### 2.2. Neuropsychological Examination

Patients first underwent a general neuropsychological examination, and then the Semantic Learning Experimental Battery.

#### 2.2.1. General Neuropsychological Assessment

The standard neuropsychological assessment included the following tests: Mini Mental State Examination [43]; Raven’s Coloured Progressive Matrices [44]; digit span [45]; Corsi block-tapping test [46]); copy and delayed recall of the Rey–Osterrieth figure B [47]; apraxia tests battery [48]; phonological and semantic fluency [45]; Luria’s motor sequences [49]; Stroop test [45]; easy naming test, word reading test, and word-to-picture matching test [12]; Laiacona’s semantic battery [50,51]; verbal and non-verbal versions of the Pyramid and Palm Trees Test (PPTT [52]); Visual Object and Space Perception test -VOSP [53].

The patients’ performances were matched to a reference group of 25 healthy controls matched for age, sex, and education collected in our lab. One-way ANOVA and post-hoc Tukey Test were used to analyse data.

Patients with mild svPPA showed a consistent pattern of selective impairment in tasks exploring semantic abilities. They showed poor performances in language tasks and struggled with naming and understanding single words, as evidenced by the “easy” naming and the “easy” picture matching tests, the errors made in semantic association tests, and the poor scores obtained in the fluency task.

Conversely, they easily passed the tasks for the VOSP battery exploring the elementary visual perception, even if some of them failed in the complex visuo-perceptual tasks (e.g., “silhouettes” and “object decision”). General abilities (MMSE) and logical-deductive reasoning (Raven’s Coloured Progressive Matrices) were preserved, and they had a good orientation in time and place. Tests exploring visuospatial skills, such as the VOSP spatial items and the copy of Rey’s complex figure, were in the normal range. Furthermore, their working and long-term memories were preserved, their performance of tasks exploring ideomotor praxis was very good, and their executive functions were almost spared.

Patients with moderate svPPA showed poorer performances in all semantic tasks.

Moreover, their performances were lower than the normal limits in several tests with linguistic mediation (for example, in the MMSE, patients struggled with providing the name of their location and the day and month of the year), whereas their performances were within the normal limits in visuospatial and visual perceptive tests. Scores obtained in praxis and visual memory tests were also normal.

Results obtained by patients with mild and moderate svPPA in the standard neuropsychological assessment are reported in Table 1.

#### 2.2.2. The Semantic Learning Experimental Battery

The battery is made up of twenty stimuli characterized by drawings of non-real (abstract) figures (Figure 1) all having similar dimensions.

The battery consists of 4 tests with the purpose of examining specific aspects of the lexical-semantic level of discourse, such as naming, i.e., the attribution of a verbal label to a concept (“Provide the novel words associated to novel abstract figures” test), the ability to recognize individual concepts through recognition tasks between stimuli belonging to the same abstract category (“Novel words to novel abstract figures (abstract distractors)” matching test), or to create semantic associations (“Matching black and white versions of the novel abstract figures to arbitrary colours” test), and the ability to distinguish individual abstract concepts from distractors formed by concrete stimuli (“Novel words to novel abstract figures (concrete distractors)” matching test).

Two versions of the stimuli are proposed: one in black and white, the other in colour. In the coloured version, the figures are divided into 4 groups: 5 red, 5 yellow, 5 blue, and 5 green coloured figures.

A neologism is associated with each figure. These neologisms consist of anagrams of Italian words that are consistent with the Italian phonological rules.

The assessment of the individual learning after training is performed at the end of the training sessions and three months after the last session via a series of tests:(1)“Provide the novel words associated to novel abstract figures” test: the individual is presented with 20 non-real figures; each figure is placed in the centre of an A4 sheet; the individual is asked to name (i.e., provide the neologism associated with) the figure. Only the answers with all phonemes produced were considered correct. The maximum score is 20.(2)“Novel words to novel abstract figures (abstract distractors)” matching test: the individual is presented with 20 sheets. Each sheet contains 5 of the abstract learned figures arranged vertically; the figures are presented in random order but each stimulus appears the same number of times as all the others. One of the 5 figures represents the target and the others are distractors. The examiner reads out the novel word and asks the individual to indicate the corresponding figure. The maximum score is 20.(3)“Matching black and white versions of novel abstract figures to arbitrary colours” test: the individual is presented with the black and white versions of the 20 novel figures. These figures are presented one at a time along with 4 coloured rectangles (red, green, yellow, and blue); the individual is asked to match the colour to the novel figure. The maximum score is 20.(4)“Novel words to novel abstract figures (concrete distractors)” matching test: the test is structurally similar to the previous test; the difference is that distractors are formed by figures of real objects. As in the previous test, the individual is asked to indicate the figure corresponding to the neologism read by the examiner.

### 2.3. Study Design

The experimental study was performed according to two modalities (formal training and home training), allowing the individuals and their relatives to choose the procedure that best fitted with adherence to the training program.

#### 2.3.1. Formal Training

The formal training was carried out in 12 sessions—two training sessions per week—lasting 45 min each.

In the first session, the individuals were asked to copy each figure and its name on a sheet in order to facilitate learning. After the 20 images were copied with their names, the latter were covered and their recall was requested. The characteristics of each single image were then analysed, and we worked on the neologism–image association through copying, immediate and delayed recall. At the end of the session, all twenty images were presented again in a random order, one after the other, and patients were asked to recall the neologism corresponding to each figure.

Starting from the second session, the treatment was organized through the following exercises:Presentation of stimuli in a fixed, predetermined order: The order of presentation was selected according to the difficulty and complexity of the neologisms and their respective figures. First the red group was presented, which mainly consisted of disyllabic and—only occasionally—trisyllabic words with the same initial letter (“soro”, “polve”, “scepa”, “venopa”, “vocalla”), then the yellow group (“neole”, “braze”, “curango”, “celtollo”, “chefrotta”) with two disyllabic and three trisyllabic words, then the blue group (“nace”, “fogu”, “bigliotta”, “giuttagra”, “tillobore”) with words of different length (two to four syllables). Finally, the green group was presented (“ganno”, “guonta”, “micacia”, “afelente”, “panilanto”): it was the most complex group, as it included two words with four syllables. During the presentation, an attempt was made to memorize the characteristics of the individual figures without asking to recall the corresponding neologism. For example, the “soro” figure was shown and the individual was invited to pay attention to the different triangles that formed this figure.Association between written targets and the corresponding abstract figures: Cards with the names of figures were given to the individual, who was asked to correctly associate each card with the corresponding image.Naming abstract figures by colour category: the individual was asked to name all the images belonging to the same colour-category presented, without the aid of cards with the respective neologisms.Matching neologisms with abstract figures belonging to the same colour category: All the images of the same colour were placed on the table and the individual was asked to indicate the figure corresponding to the verbal target.Identification of the colour category to which each abstract figure belongs: The black and white figures were presented. A sheet of paper containing four coloured rectangles (red, yellow, green, and blue) was presented as well. The individual was asked to point to the colour corresponding to each figure.

#### 2.3.2. Home Training

Patients were trained to learn novel concepts and new words with the help of their caregivers at home. Patients who lived far from our hospital or could not be taken to the hospital on the scheduled dates for the training opted for this modality. The home training was carried out by the caregiver after being instructed by the speech therapist, so that the type of exercises proposed were similar to the formal training. The program consisted of three 30-min training sessions per week for 12 weeks.

## 3. Results

### 3.1. Quantitative Analysis

#### 3.1.1. Post-Treatment Evaluation

Individual evaluation at the end of the training sessions was carried out one week after training, and Kruskal–Wallis rank-sum tests on correct responses in each task revealed group effects (3 samples: mild svPPA, moderate svPPA, controls) in three tasks: “Provide the novel words associated to novel abstract figures” test (χ2 = 15.36; df = 2; *p* < 0.001), “Novel words to novel abstract figures (abstract distractors)” matching test (χ2 = 11.92; df = 2; *p* = 0.003), and “Matching black and white versions of novel abstract figures to arbitrary colours” test (χ2 = 9.81; df = 2; *p* = 0.007). No group effect was detected in the “Novel words to novel abstract figures (concrete distractors)” matching test (χ2 = 4.5; df = 2; *p* = 0.11).

Pairwise comparisons between groups via Mann–Whitney–Wilcoxon tests on correct responses produced the following results:-− In the “Provide the novel words associated to novel abstract figures” test, patients with mild svPPA gave fewer correct responses than controls (W = 44; *p* = 0.03), and so did patients with moderate svPPA (W = 28; *p* < 0.001). Patients with moderate svPPA gave fewer correct responses than patients with mild svPPA (W = 10; *p* = 0.004; Figure 2a).-In the “Novel words to novel abstract figures (abstract distractors)” matching test, there was no difference between controls and patients with mild svPPA (W = 65.5; *p* = 0.36); however, there was a significant difference between controls and patients with moderate svPPA (W = 28; *p* < 0.001), as well as between patients with mild and moderate svPPA (W = 10; *p* = 0.004; Figure 2b).-In the “Matching black and white versions of novel abstract figures to arbitrary colours” test, no difference was detected between patients with mild svPPA and controls (W = 76; *p* = 1); however, patients with moderate svPPA gave fewer correct responses than controls (W = 28; *p* < 0.001) and fewer correct responses than patients with mild svPPA (W = 10; *p* = 0.004; Figure 2c).-In the “Novel words to novel abstract figures (concrete distractors)” matching test, there was no difference between controls and patients with mild svPPA (W = 76; *p* = 1), between controls and patients with moderate svPPA (W = 43; *p* = 0.055), and between patients with mild and moderate svPPA (W = 22; *p* = 0.57; Figure 2d).

The type of training (formal rehabilitation vs. home training) received by the patients did not influence their performances. No significant differences were found between individuals undergoing a formal rehabilitation and individuals who received home training [“Provide the novel words associated to novel abstract figures” test (W = 21; *p* = 0.46); “Novel words to novel abstract figures (abstract distractors)” matching test (W = 22; *p* = 0.57); “Matching black and white versions of novel abstract figures to arbitrary colours” test (W = 24; *p* = 0.8); “Novel abstract figure (concrete distractors)” matching test (W = 22; *p* = 0.57)].

#### 3.1.2. Three-Month Follow-Up

The evaluation was repeated three months after the end of the rehabilitation treatment. Results were calculated on 6 controls and 11 patients with SD, as one of the patients with moderate SD refused the evaluation, explaining that he was unable to remember any of the items proposed. Figure 3 shows the performance values for each of the 11 patients with semantic dementia.

Kruskal–Wallis rank-sum tests on correct responses in each task revealed group effects in the “Provide the novel words associated to novel abstract figures” test (χ2 = 15.39; df = 2; *p* < 0.001), in the “Matching black and white versions of novel abstract figures to arbitrary colours” matching test (χ2 = 8.47; df = 2; *p* = 0.014), in the “Novel words to novel abstract figures (abstract distractors)” matching test (χ2 = 10.85; df = 2; *p* = 0.004), and also in the “Novel words to novel abstract figures (concrete distractors)” matching test (χ2 = 19.86; df = 2; *p* < 0.001).

Pairwise comparisons between groups via Mann–Whitney–Wilcoxon tests on correct responses produced the following results:-In the “Provide the novel words associated to novel abstract figures” test, patients with mild svPPA gave fewer correct responses than controls (W = 39.5; *p* < 0.001), and so did patients with moderate svPPA (W = 28; *p* < 0.0.001); patients with moderate svPPA gave fewer correct responses than patients with mild svPPA (W = 10; *p* = 0.01; Figure 4a).-In the “Novel words to novel abstract figures (abstract distractors)” matching test, there was a significant difference between controls and patients with mild svPPA (W = 52; *p* = 0.03), and between controls and patients with moderate svPPA (W = 28; *p* < 0.001), as well as between patients with mild and moderate svPPA (W = 10; *p* = 0.01; Figure 4b).-In the “Matching black and white versions of novel abstract figures to arbitrary colours” test, no difference was present between patients with mild svPPA and controls (W = 63.5; *p* = 0.27); patients with moderate svPPA gave fewer correct responses than controls (W = 28; *p* < 0.001) and patients with mild svPPA (W = 10; *p* = 0.014; Figure 4c).-In the “Novel words to novel abstract figures (concrete distractors)” matching test, there was no difference between controls and patients with mild svPPA (W = 76; *p* = 1), whereas patients with moderate svPPA produced fewer correct responses than controls (W = 43, *p* = 0.009) and patients with mild svPPA (W = 10; *p* = 0.01: Figure 4d).

The type of training (formal rehabilitation vs. home training) the patients received did not influence their performances after three months. No significant differences were found between individuals undergoing a formal rehabilitation and individuals who received home training: [“Provide the novel words associated to novel abstract figures” test (W = 12; *p* = 0.28); “Novel words to novel abstract figures (abstract distractors)” matching test (W = 14; *p* = 0.49), “Matching black and white versions of novel abstract figures to arbitrary colours” test (W = 18; *p* = 1); “Novel words to novel abstract figures (concrete distractors)” matching test (W = 16; *p* = 0.77)].

#### 3.1.3. Correlation between Performance in the Experimental Battery and Language Performance

The performance of svPPA patients in this experimental battery was correlated with the performances in similar tasks that explore semantic aspects of standard language, in particular with naming test (naming task from the Laiacona’s battery [50,51]), single word comprehension (verbal comprehension within category from the Laiacona’s battery [50,51]), and semantic association tests (Pyramids and Palm Tree test verbal and visual versions [52]). Bonferroni correction was applied to these correlations (*p*-value significant was <0.008).

The “Provide the novel words associated to novel abstract figures” test correlated with the naming task of the Laiacona’s Battery (r = 0.94; *p* < 0.001), the single word comprehension from the Laiacona’s Battery (r = 0.95; *p* < 0.001), and the Pyramids and Palm tree test verbal version (r = 0.76; *p* = 0,004; Bonferroni correction). Correlation with the visual version of the Pyramids and Palm tree test was not significant (r = 0.66; *p* = 0.018) according to the Bonferroni correction.

The “Novel words to novel abstract figures (abstract distractors)” matching test correlated with the naming task of the Laiacona’s Battery (r = 0.85; *p* < 0.001) and the single word comprehension from the Laiacona’s Battery (r = 0.88; *p* < 0.001); the correlation was not significant with the Pyramids and Palm tree test.

The “Matching black and white versions of novel abstract figures to arbitrary colours” test correlated with the naming task of the Laiacona’s Battery (r = 0.83; *p* = 0.001) and the single word comprehension from the Laiacona’s Battery (r = 0.85; *p* < 0.001); no significant correlation was found with the with the Pyramids and Palm tree test.

### 3.2. Qualitative Analysis of Specific Cognitive and General Psychological Problems

Some observations made in our patients during the semantic learning study could be of interest from a cognitive point of view. A first observation was that all patients with semantic dementia used the same (visual spatial) retrieval strategy in the naming and comprehension tasks. This strategy can be inferred by the fact that in the semantic learning tasks, individuals were given series of 4 sheets with 5 geometrical figures drawn on each sheet and the relative neologism indicated next to each figure. In both the naming and comprehension tests, all individuals were capable of remembering the exact location of the items in the sheets (for example, in front of the geometric figure associated with the neologism “neole”, when the examiner asked to name it, the patient replied “This is the one in third position in the second sheet”). Each spatial location reference provided was accurate. Therefore, even patients with severe semantic dementia, despite not being able to name or recognize the target among different distractors, were able to retrieve the exact location of different items in the sheets provided.

A second observation concerned a subgroup of six patients (three in the mild phase and three in the severe phase of the disease) who were given two spontaneous recall tasks before the evaluation at the end of the treatment. All individuals were first asked to spontaneously recall and write the names of the neologisms, and then draw the new figures they learned. In the three individuals with mild svPPA, the number of neologisms and novel abstract figures they could independently write and draw in the free recall tasks was almost comparable to the number of abstract pictures named. Conversely, in individuals with moderate svPPA, the number of neologisms written and novel abstract figures separately drawn in the free recall test was greater than the number of novel abstract figures correctly named. One of the patients commented on his performances as follows: “I remember some names and figures, but I can’t tell which figure corresponds to a given name and which name to a given figure” (see Figure 5).

## 4. Discussion

Results of previous investigations assessing the influence of learning capacity on language rehabilitation in svPPA patients [26,27,28,30,31,32,39,40] often assumed that improved performances might be due to the interactions existing in svPPA between episodic and semantic memory, because empirical (e.g., [12]) and theoretical reasons (e.g., [54] suggested that an interdependence between these memory systems should exist. Consistent but anecdotal observations were reported on this topic by Snowden et al. [55,56], Graham et al. [28], and Hodges and Graham [57]. Snowden et al. [55] showed in a first paper that autobiographical experience has an important role in influencing information preservation in svPPA. Furthermore, during the investigation of celebrity knowledge by svPPA patients, these same authors [56] revealed that performances are better for contemporary (recent) than for past (remote) information. These findings are consistent with explanations of the “temporal gradient” effect of retrograde amnesia, indicating a bidirectional interaction between autobiographic and semantic memory. Similar conclusions were reached by Graham et al. [28] and Hodges and Graham [57], who reported cases of svPPA patients who could show preservation of new learning despite their loss of semantic knowledge, and showed that this new learning capacity heavily relies upon intact perceptual processes and stimulus familiarity. However, an influence of learning capacity on responses to aphasia rehabilitation has also been documented by authors who have investigated novel word learning in people with stroke aphasia (e.g., [29,33,34,35,36,37]), showing that adults with chronic post-stroke aphasia are able to learn novel words.

These results could suggest that improved performances observed after word learning by svPPA patients could be due to a reorganization of semantic memory, rather than to an interaction between episodic and semantic memory systems. In any case, the ability of svPPA patients to learn new concepts has never been explicitly addressed, nor can it be inferred from the patients’ clinical history. While normal individuals learn new items day by day in episodic memory (and the clinical history of AD patients shows they are no longer able to acquire new information), svPPA patients are much less frequently exposed to new concepts, thus allowing us to understand if they are or are not able to learn these new concepts with some efficiency. Results of investigations conducted on aphasic stroke patients suggest that this is possible, but a clear response about the underlying mechanisms has never been formulated, because many different semantic, non-semantic, representational, and control mechanisms can influence the results obtained from these patients [8,58,59,60].

The main purpose of the present study, therefore, consisted in trying to evaluate if semantic learning could support the hypothesis that assumes that the loss of knowledge typical of svPPA is not simply due to a loss of information stored in semantic memory, but to a more general defect of semantic acquisition, storage, and retrieval. In order to check this hypothesis, we took into account the capacity of the semantic system to learn both new concepts and new word forms. For this reason, we assessed the semantic learning capacity of svPPA patients using a methodology similar to the one employed by Digman et al. [36], studying the novel word learning of aphasic stroke patients by pairing novel word forms with new semantic representations.

We also assumed that the abstract vs. concrete nature of the stimuli and the arbitrary nature of the links existing between new conceptual representations and novel word forms should increase the difficulties met in learning new semantic information. Therefore, our “Semantic Learning Experimental Battery” included both a task at least partially based on concrete material—the “Novel words to novel abstract figures (concrete distractors)” matching test—and tasks based on abstract figures associated by arbitrary links, such as the “Provide the novel words associated to novel abstract figures” test, the “Novel words to novel abstract figures (abstract distractors)” matching test, and the “Matching black and white versions of novel abstract figures to arbitrary colours” test.

The main results obtained with this methodology were consistent with our predictions, because the lowest scores in the semantic learning tasks were found (at the end of the training sessions and at the three-month follow-up) in the svPPA patients with the most advanced forms of this disease. In addition, the size of the difference between controls and patients with mild and moderate svPPA was greater in the matching tasks based on the arbitrary association between novel words and novel abstract figures than in the “Novel words to novel abstract figures (concrete distractors)” matching test. The strong correlations found between levels of semantic loss and new semantic learning strongly suggest that the loss of semantic information observed in svPPA is due to a general disruption of mechanisms concerning both learning and storing of semantic knowledge.

Moreover, the existence of a strong relationship between loss of semantic knowledge and disruption of semantic learning was confirmed by the significant correlations found in svPPA patients between scores obtained in the semantic learning tasks and those obtained in the Laiacona et al. [49,50,51] battery for the assessment of semantic memory disorders and in the verbal version of the Pyramids and Palm tree test [52].

Some additional information about the mechanism underlying the learning defect of our SD patients could be provided by a qualitative analysis of some observations made during the semantic learning study. A first observation was that their learning defect was not global, but restricted to the semantic domain, as in both the naming and in the matching tasks, these individuals could remember the exact location of the items in the sheets and offered this preserved ability as a proof that they were not “demented.” A second observation was that the greatest difficulty of patients in more advanced stages of svPPA seemed to consist of an inability to correctly associate novel abstract figures to novel arbitrary word forms. These patients, indeed, were able to write down some new names or draw some new abstract figures, but could not associate the figures to the corresponding names, as one of them explicitly noted.

In conclusion, the present study demonstrates that in svPPA, the acquisition of new concepts is still possible, although the individual ability varies according to the magnitude of the semantic impairment. This evidence, if confirmed by future research, can also have important implications from a clinical point of view. In a disease where there is currently no possibility of remission, cognitive rehabilitation could contribute to enhance residual cognitive functions.

## 5. Conclusions

This study provided two results of some interest within the theoretical framework guiding our investigation. The first outcome lies in a comparative analysis of results obtained across the different tasks forming the “Semantic Learning Experimental Battery.” The second consists of a correlative analysis of results obtained by svPPA patients in the semantic learning battery and in neuropsychological tasks assessing the disruption of semantic knowledge. The first results show that the learning capacities of svPPA patients adhere to the typical developmental lines of the semantic system, thus suggesting that the underlying learning process may develop within the semantic rather than the episodic memory system. The second results show there is a strong relationship between loss of semantic knowledge and disruption of semantic learning, thus suggesting that a general defect of semantic acquisition, storage, and preservation may be responsible for the loss of knowledge observed in svPPA patients. Obviously, we are aware that our investigation lacked some extent of theoretical sophistication and methodological control. We think, however, that our results were interesting enough to stimulate a further exploration of this fascinating aspect of the human cognitive functions.

## Figures and Tables

**Figure 1 brainsci-13-00788-f001:**
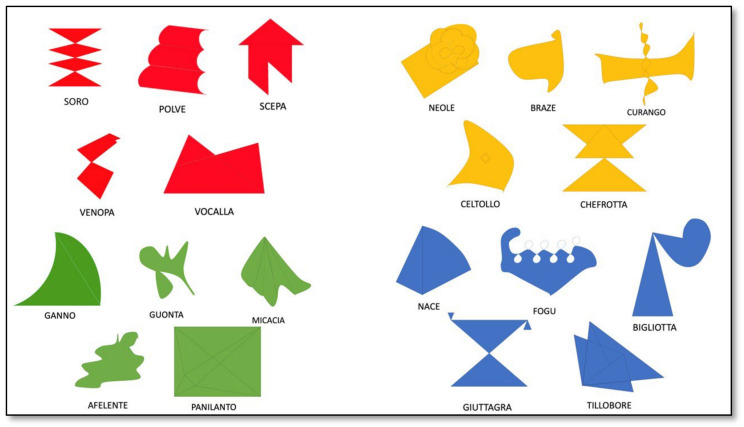
The abstract coloured figures and the corresponding neologisms.

**Figure 2 brainsci-13-00788-f002:**
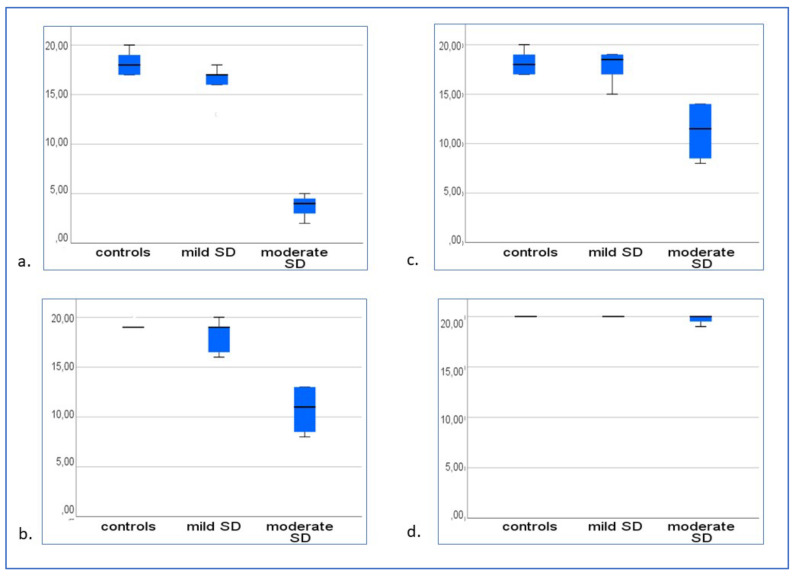
Post-treatment evaluation: (**a**) “Provide the novel words associated to novel abstract figures” test; (**b**) “Novel words to novel abstract figures (abstract distractors)” matching test; (**c**) “Matching black and white versions of novel abstract figures to arbitrary colours” test; (**d**) “Novel words to novel abstract figures (concrete distractors)” matching test.

**Figure 3 brainsci-13-00788-f003:**
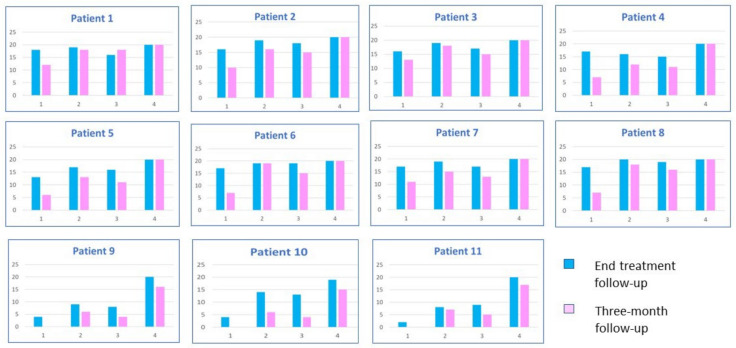
Performance of each SD patient just after rehabilitation and at the three-month follow-up. Keys: 1 = “Provide the novel words associated to novel abstract figures” test; 2 = “Novel words to novel abstract figures (abstract distractors)” matching test; 3 = “Matching black and white versions of the novel abstract figures to arbitrary colours” test; 4 = “Novel words to novel abstract figure (concrete distractors)” matching test. Patients 1–8 = mild svPPA; patients 9–11 = moderate svPPA.

**Figure 4 brainsci-13-00788-f004:**
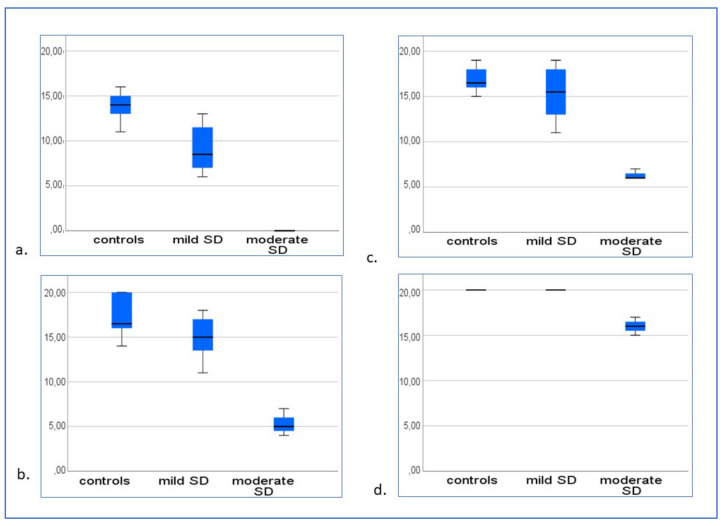
Three-month follow-up: (**a**) “Provide the novel words associated to novel abstract figures” test; (**b**) “Novel words to novel abstract figures (abstract distractors)” matching test; (**c**) “Matching black and white versions of novel abstract figures to arbitrary colours” test; (**d**) “Novel words to novel abstract figures (concrete distractors)” matching test.

**Figure 5 brainsci-13-00788-f005:**
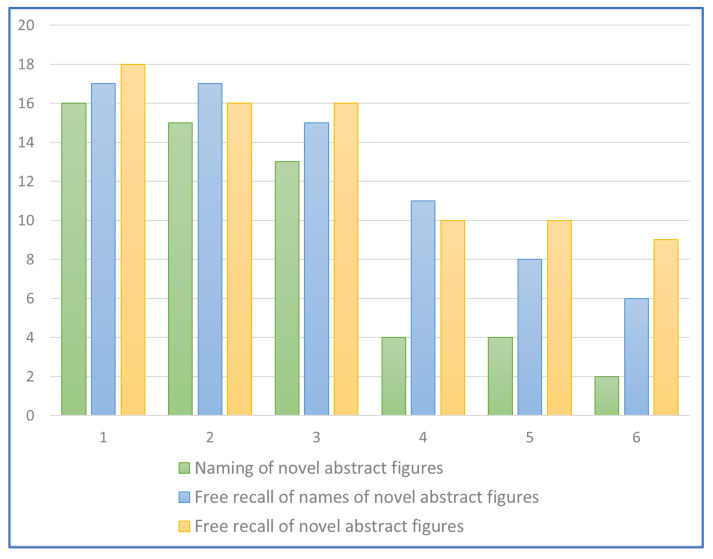
Performance of patients with mild (patients 1, 2, 3) and moderate (patients 4, 5, 6) SD in naming vs. independent free recall of novel words and novel abstract figures.

**Table 1 brainsci-13-00788-t001:** General neuropsychology.

Test (Maximum Score)	Mild SD	Moderate SD
**General Abilities**		
MMSE (30)	27 (0.02)	18.25 (2.98) *
**Memory**		
Digit span	5.25 (0.47)	4.5 (0.57)
Corsi blocks	5.62 (0.46)	5.25 (0.5)
Rey figure B:delayed recall (31)	23.87 (1.8)	20.5 (1.91)
**Perceptuo-spatial skills**		
VOSP battery		
-Shape detection test (20)	19.87 (0.35)	19.5 (1)
-Incomplete letters (20)	17.3 (2.5)	14.5 (3.8) *
-Silhouettes (30)	9.5 (2.67) *	0 (0) *
-Object decision (20)	15.87 (1.45)	10 (2.94) *
-Dot counting (10)	10 (0)	9.87 (0.35)
-Number location (10)	9.1 (0.8)	8.6 (2.3)
-Position discrimination (20)	19.75 (0.46)	19 (0.81)
-Cube analysis (10)	9.5 (0.75)	8.75 (1.5)
**Ideomotor praxis**		
Left upper limb (20)	20 (0)	20 (0)
Right upper limb (20)	20 (0)	20 (0)
**Constructional praxis**		
Rey Figure B Copy (31)	30.87 (0.35)	30.5 (0.57)
**Executive functions**		
Raven’s Coloured Progressive Matrices (36)	28.25 (2.1)	21.25 (2.75) *
Letter Fluency (F,A,S)	39.62 (14.17)	6.75 (4.11) *
Luria’s motor sequences (50)	49.5 (7.56)	43 (2.51) *
Stroop test (time first part)	34.37 (6.63)	42.25 (3.86)
Stroop test (time second part)	57.25 (8.64)	-
Stroop test (errors first part)	0(0)	0 (0)
Stroop test (errors second part)	5.8 (4.8)	-
**Language**		
Verbal fluency (3 categories 1 min each)	19.9 (9.4) *	2.5 (1.04) *
‘‘Easy’’ picture naming (40)	27.37 (4.3)	1.75 (1.7) *
‘‘Easy’’ picture reading (40)	39.87 (0.35)	36.5 (1.29) *
‘‘Easy’’ word-picture matching (40)	37.62 (1.76) *	18.75 (2.98) *
Laiacona’s Battery		
-Naming (80)	43.25(5) *	3.25 (2.62) *
-Verbal comprehension (within category) (80)	67.25 (2.9) *	(4.9) *
-Verbal comprehension (inter-category)(80)	77(1.19)	54.75 (8.99)*
PPTT (verbal version) (30)	19.87 (3.9) *	8.5 (6.35) *
PPTT (non-verbal version) (30)	27.25 (3.45)	18.5(6.65) *

* Abnormal performances (patients’ scores were compared to a group of healthy controls matched for age, sex, and education collected in our laboratory, and abnormal performances were inferior to the cut-offs of healthy individuals).

## Data Availability

The data presented in this study are available on request from the corresponding authors.

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
