# Peer review of "The Dynamic Interplay between Loss of Semantic Memory and Semantic Learning Capacity: Insight from Neologisms Learning in Semantic Variant Primary Progressive Aphasia"

_brainsci, 2023, doi:10.3390/brainsci13050788_

Round 1

Reviewer 1 Report (Previous Reviewer 2)

Page 2, lines 47-49:
“Apraxia of speech and agrammatism in language production are the core features of the Non-Fluent variant” Only one feature is required by the criteria, so this should be “Apraxia of speech and/or agrammatism in language production are the core features of the Non-Fluent variant’

“whereas phonological errors and impaired repetition of sentences are the key features of the Logopenic PPA”  The core features are “Impaired single-word retrieval in spontaneous speech and naming” and “Impaired repetition of sentences and phrases.” “Phonological errors” is a secondary criterion.

Page 2, lines 49-51: “the significant word-finding problems potentially found in the Logopenic variant are due to lexical retrieval, and not to properly semantic problems” This may be true initially, but may not be true at later stages of the syndrome. As lvPPA progresses, atrophy spreads to temporal areas associated with lexical-semantic and conceptual processing (Leyton et al., 2016), and semantic impairment may appear in lvPPA (Roncero et al., 2020).

Page 3, lines 129-130: “Predominant anterior left temporal atrophy detected by an MRI and/or pre- dominant anterior left temporal hypoperfusion at the FDG-PET scan;” Were any potential participants excluded because they had anterior temporal atrophy or hypoperfusion that was comparable across the hemispheres? If so, the text should acknowledge that this criterion is more strict than Gorno-Tempini et al. (2011), which only requires “Predominant anterior temporal lobe atrophy.”

Page 4, line 159: “(t=0.01; p=0.58).” Is this correct? The p-value would be close to 1 if t was close to zero.

Page 6, lines 226-229: For the naming response, was perfect production of the neologism required (i.e., all phonemes produced)?

Author Response

Reviewer 2 Report (Previous Reviewer 1)

The authors did some changes that improved the quality of the paper. However, a lot of the changes were rather superficial given the time restrictictions posed in a fast publication process. The paper lacks a sophisticated theoretical basis and a sound methodological approach (see all comments in previous review reports). 

The paper needs some further minor editing regarding English.

Author Response

This manuscript is a resubmission of an earlier submission. The following is a list of the peer review reports and author responses from that submission.

Round 1

Reviewer 1 Report

Summary and general evaluation

The aim of the paper is to show that semantic dementia (SD)/semantic variant of Primary Progressive Aphasia (sv-PPA) is not characterized just by loss of knowledge stored in semantic memory, but is also due to a general disruption of mechanisms allowing acquisition, storage and retrieval of semantic memories. To evaluate this hypothesis, the authors administered to 12 SD patients and 6 healthy controls a battery of semantic learning tasks, in which participants were trained to learn new conceptual representations and new word forms and to associate the former with the latter. The authors report a strong relation between loss of semantic knowledge and disruption of semantic learning: (a) the lowest scores on the semantic learning tasks were obtained by patients with moderate forms of SD; (b) significant correlations were found in SD patients between scores obtained on the semantic learning tasks and those obtained on semantic memory disorders.

The paper puts forward an interesting claim that could potentially deepen our understanding of linguistic deficits in SD and their relation to other cognitive impairments. However, there are a number of shortcomings that render the paper not publishable in its current form. Below I provide specific comments for improvement of different sections of the paper.

General comments

1) Ι found the literature review and the context within the study is placed quite poor. The authors should place their study within the broader context of linguistic deficits in SD/sv-PPA and their relations to other cognitive impairments .

2) The authors should state from the very beginning of the paper that the term SD and sv-PPA refer to the same neurodegenerative condition and should discuss sv-PPA in relation to the other variants of PPA (and not just in relation to Alzheimer’s disease), as there is considerable literature in this domain.

3) Methodology: I do not understand why the control group of the normal subjects was not individually matched to the experimental group, i.e., why the group of normal controls included just 6 participants. Individual matching is a standard practice in experimental studies within aphasiology and in general within psycholinguistic and neurolinguistic research. It is also not clear to me what the purpose of the “reference group” of 25 healthy controls, mentioned on p. 3 line 127, is. Did the authors include two controls groups in their study? What is the reasoning for such decision? Also, it is reported that the 6 control subjects were matched on age, sex and education and that demographic variables did not significantly differed between the 12 individuals with sv-PPA and the 6 control subjects (p. 3, lines 110-111). However, it is not clear on p. 3, lines 127-130 whether the demographic variables of the reference group and the sv-PPA group did not significantly differ. If I understand correctly the authors used a control group of 25 healthy individuals matched on 3 demographic variables to the experimental group to compare results of the general neuropsychological battery and another control group of 6 healthy individuals to compare results of the training/intervention study. What is the reasoning behind this decision?

4) In section 2.2.1. The Semantic Learning Experimental Battery, the authors should add what is the purpose of each of the 4 tests used in the SLE battery, which aspect of semantic processing they target, and what the predictions for the patients’ performance in each of these tests is.

5) Section 3.2, p. 10-11: I do not see what the point of a qualitative analysis is unless the authors embed their study within a clinical frame that will consider the effects of semantic learning and enhancement for the patients’ communicative abilities and quality of life. The authors make an attempt in the last paragraph of the Discussion section to discuss the clinical implications of their study but this is too short and superficial, and should be embedded and discussed within a more specific clinical context.

6) The Discussion should be rewritten as in its current form it is merely a description of the results. The discussion should also include the authors’ views on the ramifications that their findings have for hypotheses and theories regarding the underlying impairment in PPA and specifically in sv-PPA.

7) My suggestion is that the whole text must be proofread by a native speaker as there are a number of issues regarding language. Nonetheless, proofreading of the Discussion is mandatory because this section is very weak not only in terms of its content but also in terms of the language used.

Specific comments

1) Please, replace the term “aphasic” with “people with aphasia” or “individuals with aphasia” or “patients with aphasia” throughout the text. Also replace the phrase “aphasic stroke patients” with “patients with post stroke aphasia” (p. 2, line 63).

2) p. 9, lines 302-305: Provide the statistics for the effect of the type of training (formal rehabilitation vs. home training) in the after three months assessment.

Author Response

Pease see the attachment

Reviewer 2 Report

Participants: Since the 12 patients all have svPPA, the title and abstract should refer to svPPA rather than Semantic Dementia. As defined by the Neary et al. (1998) FTLD criteria, Semantic Dementia is a broader diagnosis than svPPA. Semantic Dementia can include patients who primarily have difficulty recognizing objects (associative agnosia) and/or familiar faces (prosopagnosia).

Page 2, lines 87-88: “Predominant left temporal atrophy detected by the MRI and/or predominant left temporal hypoperfusion at the FDG-PET scan” was required for patients. This requirement is different from the Gorno-Tempini et al. (2011) criteria for svPPA, which specify “predominant anterior temporal lobe atrophy,” hypoperfusion, or hypometabolism. Why was left temporal atrophy or hypoperfusion required for inclusion, rather than anterior temporal atrophy/hypoperfusion?

Page 3, line 113: “(t=0.01; p=0.58).” Is this correct?

Page 4, lines 145-147: This sentence is unclear. Was the performance of the moderate patients “worse” or “better” compared to mild patients, or compared to other types of tasks?

Page 4, lines 158-161: For the naming response, was perfect production of the neologism required (i.e., all phonemes produced)?

Page 4, line 173: The name “Novel-words to figures of real objects matching test” is not clear. As in task number 2, the participant is attempting to match a novel word to a novel abstract figure.  The difference between the two tasks is the type of distractor, so it would be more clear to name the two tasks something like:
2) Novel-words to novel abstract figures matching test (abstract distractors)
4) Novel-words to novel abstract figures matching test (real distractors)

Figures 2 and 4 - The four subfigures within each figure should all have the same scale (0 to 20).

Round 2

Reviewer 1 Report

The authors made some changes in the manuscript but given the very restricted time they had to revise it, the changes were quite superficial and not well-written. For example, in the introduction they refer to the distinction between access and storage disorders, but still the discussion is not sophisticated and not properly addressed in the Discussion section.

The authors mention that the battery of neuropsychological tests in the background is not formally part of the experimental battery and has the role of characterizing the cognitive profile of the patients for diagnostic purposes. Exactly for this reason such data (i.e., baseline data from a neuropsychological battery) should be reported in a study that investigates a neurodegenerative disease. The authors also state that “Comparison of the two groups helps to detect deficits in specific neuropsychological domain and contribute to the diagnosis.” but they do not present the data that would illustrate such comparison.

Point 5 in the original review has not been properly addressed: 5) Section 3.2, p. 10-11: I do not see what the point of a qualitative analysis is unless the authors embed their study within a clinical frame that will consider the effects of semantic learning and enhancement for the patients’ communicative abilities and quality of life. The authors make an attempt in the last paragraph of the Discussion section to discuss the clinical implications of their study but this is too short and superficial, and should be embedded and discussed within a more specific clinical context.

To sum up, the paper has theoretical and methodological flows that render it not publishable in its current form.

Last, serious proofreading is necessary for publication in a peer-reviewed journal.
